# Pollen Germination and Pollen Tube Growth in Gymnosperms

**DOI:** 10.3390/plants10071301

**Published:** 2021-06-26

**Authors:** Maria Breygina, Ekaterina Klimenko, Olga Schekaleva

**Affiliations:** Department of Plant Physiology, Biological Faculty, Lomonosov Moscow State University, 119991 Moscow, Russia; kleo80@yandex.ru (E.K.); anny.shirly.ganbatte@gmail.com (O.S.)

**Keywords:** plant reproduction, pollen germination, pollen tube growth, gymnosperms, conifers, *Picea*, *Pinus*, male gametophyte, polar growth

## Abstract

Pollen germination and pollen tube growth are common to all seed plants, but these processes first developed in gymnosperms and still serve for their successful sexual reproduction. The main body of data on the reproductive physiology, however, was obtained on flowering plants, and one should be careful to extrapolate the discovered patterns to gymnosperms. In recent years, physiological studies of coniferous pollen have been increasing, and both the features of this group and the similarities with flowering plants have already been identified. The main part of the review is devoted to physiological studies carried out on conifer pollen. The main properties and diversity of pollen grains and pollination strategies in gymnosperms are described.

## 1. Introduction

Gymnosperms are amazing representatives of the flora. On the one hand, they are ancient plants with primitive characteristics of anatomical structure; on the other hand, they are perfectly adapted to their habitat and are the dominant species in many ecosystems due to their impressive size and longevity, with their reproductive system being of particular interest. It has progressive features, because in this group the reduced male gametophyte—pollen grain—first appeared, as well as the ability to form seeds. In addition, this group still represents a wide variety of reproductive patterns, strategies, and relationships. For example, the degree of gametophyte reduction varies, there are both zooidogamy and siphonogamy, and the reproductive process can both be relatively fast and last over several years. Botanical and evolutionary aspects, however, have already been described in sufficient detail in other reviews [1,2,3,4], whereas here we focus on the physiological data accumulated during the study of pollen germination in conifers; an informative overview of this topic was published in 2005 [5]. Here, we mainly consider new data accumulated over the past 16 years, as well as some important aspects that were outside the focus of the previous review.

## 2. Features of Gymnosperm Pollen Grains

The transition from zooidogamy to siphonogamy, that is, to fertilization by immobile gametes instead of motile spermatozoa, occurred during the evolution of gymnosperms, and nowadays both options are found in this taxon [1]. Extant gymnosperms include 12 main families and 83 genera. The term “conifers” refers to a group of gymnosperms that make up the division Pinophyta. To date, physiological studies of gymnosperms were carried out almost exclusively on conifers (mainly pine and spruce, rarely cypress, fir, and others), and all of them are characterized by typical siphonogamy, that is, fertilization with immotile gametes. The physiology of conifer pollen is discussed in the following sections.

The male gametophyte of gymnosperms is generally more complex than that of flowering plants; it develops and germinates more slowly; and it is formed through three to five mitoses, which in different groups taxa occur at different stages—before or after pollination [6]. In this regard, pollen released from microsporangia consists of different number of cells (Figure 1): four cells in pine and ginkgo (two prothallial cells, antheridial cell and tube cell), two in sequoia, and one in juniper [6,7,8]. Male gametes in all cases are formed after pollination.

Pollen grain morphology in gymnosperms is very diverse, as well as their structure, but several key patterns can be distinguished that are typical for cycads, ginkgo, gnetales, and conifers (Figure 2).

Gametophyte cells are separated by thin walls, and from outside, as in flowering plants, are protected by inner cellulosic wall (the intine) and massive outer sporopollenin wall (exine) [11]. Apertures are specialized areas, where the exine is thinner and the intine is usually thickened, channeled, or forms multiple layers [12]. Pollen wall in many cases has polar structure, and the polarity of microspores of gymnosperms, such as angiosperms, is determined during meiosis [8]. The proximal pole (the one closer to the tetrad center) later becomes more convex, and the distal pole becomes flatter, for example, in *Ginkgo biloba* (Figure 2b); exine is thicker on the proximal side, and the large aperture is located on the distal side. Cytological polarity is manifested in the cell arrangement: prothallial cells are located at the proximal end, and the large tube cell is located at the distal end [13,14,15]. In cycads, the aperture occupies almost half of the pollen grain surface, exine is the thickest in the proximal region and the thinnest in the distal one, intine is relatively thin, and exine is smooth (Figure 2a) [16]. The question of the number of apertures and polarity in Gnetales has not yet been fully clarified. In *Gnetum*, pollen is apertureless with internal polarity, and exine is embossed (Figure 2d) [10,17,18].

In most Pinaceae plants, pollen is saccate, which means, it has *sacci* (air sacs)—large hollow outgrowths formed by the exine (Figure 1 and Figure 2c). The sacs are located at the distal end of the pollen grain, and the exine of the proximal part is thickened. The presence of air sacs is associated with the pollination mechanism in many conifers: pollen landing on a pollination drop. This is the liquid secretion of the receptive ovule, which forms a large drop and rises to the micropyle surface. The drop holds pollen and pulls it into the ovule, where it germinates [19,20]. As it was shown in a model experiment, the floating of pollen grains in pollinating droplets (due to the air sacs) is an effective mechanism for their retraction into micropyle; therefore, air sacs serve not only for pollen transfer by the wind [21]. The number of air sacs, as well as the size of pollen grains and the number of mitoses during the gametophyte development, are variable [8]. Pollen of gymnosperms, unlike angiosperms, are not covered with a sticky lipid-rich pollenkitt [6].

## 3. Pollination and Pollen Germination in Gymnosperm Species

Approximately 98% of gymnosperm plant species are wind-pollinated [22], which largely determines their structural features (for example, specific shape of pollen grains and air sacs) and physiology. It is known that pollen of almost all modern gymnosperms, with the exception of some species of Araucariaceae and *Gnetum*, is characterized by a high degree of dehydration [23] and can cover great distances. Pollination with insects (mainly beetles) has been described in some Cycadales and Gnetales species [17,24], including fossils, which became possible due to the discovery of coprolites in the cones of ancient cycads [25]. Extant pollen grains have significant differences in wall structure, depending on whether they are carried by the wind or by insects, even within the same genus, which was shown for *Ephedra* [17].

When cultivated in vitro, pollen of many coniferous plants germinates during 1–2 days. Protocols have been developed for many species [26,27,28], allowing efficient pollen germination and monitoring of its behavior for a long time. The studies of conifer pollen tubes in vitro revealed their differences from the tubes of angiosperms, including cytoskeleton organization, regulation of organelle movement, and endo/exocytosis [5], which we will discuss in the next section.

The place on the grain surface where the pollen tube appears in gymnosperms is not predetermined to the same extent as in most flowering plants. The latter in most cases have one of several apertures, intended for the fast tube outlet [29]. According to data obtained on *Arabidopsis*, the optimal number of apertures is three [30]. In saccate conifer pollen grains, the pollen tube emerges between the sacs, at the pole opposite to prothallial cells. Aperture area in this case is a furrow with relatively thin exine, and in its distal areas the appearance of exine ruptures is most likely (Figure 3a,c) [31]. However, since there are two such areas, some of the pollen grains produce not one but two tubes (Figure 3a), which could be considered as a feature that reduces the germination rate, or as a potential adaptation. In a study conducted on pollen grains of blue spruce germinating in vitro, this phenomenon was described as “bipolar germination”, and it was found to occur only in case of optimal germination medium, and, apparently, it is not an adaptation for obtaining additional nutrients [31]. If there are no air sacs in the pollen grain, but it is polarized, then the place where the tube exits is a rather wide zone opposite to prothallial cells (*Ephedra* or *Welwitschia* pollen) (Figure 3b,d) [32]. No signs of polarity were found in unicellular pollen (juniper), and it is believed that the place of tube appearance is not determined [6,33,34].

In most gymnosperms, a pollen grain upon reaching the female cone lands on a pollination drop. Droplet volume ranges from 10 nL to 1 μL [36]; large drops are visible to the naked eye [17]. The list of species that do not have a pollination drop reduces as pollination is being studied closely, including through phylogenetic mapping [4]. A small or rapidly disappearing drop is often found. A pollination drop is, first of all, an apoplastic liquid. It contains inorganic substances, carbohydrates, and proteins, including enzymes [20,33,36,37]. Comparison of the pollination drop with the more studied apoplastic fluids of angiosperms (stigmatic and ovular exudates, nectar) revealed a significant similarity in their composition [20]. On this basis, it was suggested that the droplet functions are not limited to delivering pollen to the ovule [20]. Droplet enzymes, in particular chitinases, may be involved in protecting the ovule from pathogens. Possible regulatory functions of droplet components during pollen germination are also discussed. However, all these assumptions have not yet been verified experimentally.

Different pollination strategies are found in conifers [2,38]. Saccate pollen of *Pinus* and *Picea* floats on the surface of the pollination drop and with it is transferred to the nucellus, where it germinates. In *Abies*, an analogue of a pollination drop is formed from moisture collected after rain or dew. Pollen often germinates in the micropilar canal and the tube grows towards the nucellus. In *Larix*, the pollination drop is absent, but the micropilar canal is filled with ovular secretion. Surrounded by this secretion, pollen hydrates, swells, and sheds exine. In this form, it floats to the nucellus, where forms a pollen tube. This usually happens a few weeks after pollination. In some *Tsuga* species and all Araucariaceae species, there is no pollination drop [4]—pollen lands on the conical surface next to the ovule and may remain there for several weeks. It then germinates and forms a long tube that grows through the micropyle into the ovule and reaches the nucellus. The variety of patterns of pollen behavior in the female cone is extensively discussed in the literature, since it can provide important information about the evolution of pollination [2,20,22].

Сomparing angiosperms and gymnosperms, we find that the male gametophyte of the latter often has to cover a shorter distance after pollination, but it does it for a much longer time. Male gametophyte germination and growth occur slowly at all stages: the hydration of conifer pollen usually occurs in the first day after pollination, and pollen tube appears within a few days, while in flowering plants these processes take minutes and hours [5,39]. Thus, growth rate of *Picea abies* pollen tube is about 20 µm/h, which is a striking contrast compared to 300–1500 µm/h in angiosperms. In addition, conifers are characterized by a period of long dormancy when the pollen tube does not grow. In particular, the tube growth can stop for the time required to complete female gametophyte development—up to a year. In this case, a few days before fertilization, the pollen tube resumes its growth and delivers sperms to the ovule. For example, in pine and some Araucariaceae, pollen germinates shortly after pollination and the tube enters the nucellus. The dormant period lasts from mid-summer to the next spring. In *Pinus taeda*, the dormant period of the tube starts when meiosis begins in the ovule and ends a few days before fertilization [8]. Thus, for different plant groups, the time from pollination to fertilization varies from several weeks (most species of Cupressaceae and Pinaceae) to a year (*Pinus* and some Araucariaceae) [8].

## 4. Structural Network in Conifer Pollen Tubes

### 4.1. Cytoskeleton and Organelle Movement

There are no callose plugs in pollen tubes of conifers, and thus the entire tube is filled with a single array of cytoplasm and is cytologically subdivided into apex and body (Figure 4). Axial bundles of microtubules and microfilaments, the generative cell, the nucleus of the tube cell, numerous amyloplasts, vacuoles, and other organelles are localized in the body. The clear zone, free of amyloplasts, but enriched in mitochondria and endomembrane components, extends 20–30 µm from the tip. This area (apex) can be easily distinguished from the rest of the tube (body). The accumulation of vesicles in the pollen tube apex looks like a crescent [40]. Organelle movement in *Pinus sylvestris* and *Picea abies* pollen tubes was described as a direct fountain, not reverse, as in angiosperms [41]. Organelles move forward to the tip along the central axis and backward along the cell periphery [42]. The different direction of organelle movement in pollen tubes of angiosperms and gymnosperms is associated with the opposite polarity of actin microfilaments [40,43]. There are no data on the orientation of microfilaments in gymnosperm tubes, but the mathematical model confirms this version [42].

Pollen tube growth regulation was studied in greatest details in *Picea* [44,45,46,47,48]. Microfilaments’ destruction completely stopped the tube growth. At the same time, vesicle movement at the tip was disorganized and, as a result, apical cell wall construction was disrupted [46,47]. However, destruction of microtubules also blocked pollen tube growth, disrupting organelle movement at the tip and actin organization [40,48,49]. This effect reveals an important difference between pollen tubes of conifers and angiosperms, in which microtubules transport the male germ unit while the long-distance transport of other cytoplasmic structures mainly relies on the actin cytoskeleton [43,50,51,52]. Moreover, such an experimental effect switched the direction of organelle movement: a reverse fountain pattern, characteristic of angiosperms, appeared in spruce [40]. Notably, a similar effect was observed in experiments with modulation of intracellular [Ca^2+^] in pollen tubes [53]. It can be concluded that organelle movement in conifer pollen tubes is provided by both microtubules and actin microfilaments [48], which, in turn, are controlled by Ca^2+^ ions (Figure 5). The interaction between actin and tubulin cytoskeletons and their functional connection can be illustrated by mitochondrial movement in *Picea wilsonii* pollen tubes [49]. Actin and myosin directly move mitochondria, and microtubule dynamics, affecting actin organization, thereby controls the speed, trajectory, and location of mitochondria. Disruption of the actin cytoskeleton also affected proteome of cultivated Meyer’s spruce (*Picea meyeri*) pollen—more than 80 differentially accumulated proteins were identified (out of a total of 600) and functionally grouped into the following categories: signaling, cytoskeleton-associated, cell growth, and carbohydrate metabolism [54]. Latrunculin treatment greatly altered the morphology of Golgi apparatus, mitochondria, and amyloplasts. Disturbances in the organelle structure were combined with differential expression of proteins involved in their functioning. In this study, complex consequences of actin cytoskeleton destruction in pollen tubes, including global changes in the protein profile and cellular cytoarchitecture, were revealed [54].

During pollen tube growth polar distribution of ion transporters, ROS and organelles are maintained. Mitochondria and other organelles move along the actin bundles (orange) in a fountain pattern. Calcium gradient in the cytoplasm with the highest at the tip (red staining), pH gradient (blue staining with alkaline values at the tip), ROS apical gradient (H_2_O_2_ molecules are shown as asterisks), and membrane potential gradient (membrane staining with red marking depolarized apical compartment) are characteristic for growing tubes. Mitochondria produce superoxide radical. Calcium signaling is shown: kinesin-like calmodulin-binding protein (KCBP) binds to calmodulin (CaM) and regulates intracellular motility in a Ca^2+^-dependent manner.

### 4.2. Cell Wall

In pollen tubes of coniferous plants, the pattern of cell wall deposition differs markedly from that of flowering plants, although there are similarities [5,45]. Thus, in pine, callose is found not in the basal part, but, on the contrary, at the apex of the pollen tube and in the relatively young part, disappearing closer to the grain [55]. Later, approximate composition of pollen tube walls was studied in 14 gymnosperm species using cytochemical staining and monoclonal antibodies. Despite small differences, the pollen tubes of Podocarpaceae, Pinaceae, Taxodiaceae, and Cupressaceae (Coniferales) had similar components of the cell wall: a large amount of cellulose and arabinogalactan proteins were present in the tube wall, while almost in all cases there were few pectins [56]. Only *Cycas revoluta* pollen wall was found to contain a lot of pectins, while *Ginkgo biloba* had a large amount of β-(1,3) (1,4)-glucan (mixed-linkage glycan) in the tube wall. Callose was clearly shown only in *Podocarpus nagi* and *Chamaecyparis obtusa* pollen tubes (using aniline blue), and *Cryptomeria japonica* (using monoclonal antibodies) [56]. In sum, all studied gymnosperm pollen tubes had walls arranged differently than those typical for angiosperm pollen tubes, which have a pectin-rich apex and a shank part made up mainly of callose-rich inner layer and pectin outer layer [56]. These results suggest that pollen tube wall composition may reflect the taxonomic relationship between gymnosperms as well as significant differences in their type and growth rate from flowering plants.

In angiosperms, among the main components that determine the cell wall rigidity are pectins, which, as mentioned above, are part of the apical and distal parts of the pollen tube wall [57]. On this point, gymnosperms also have significant differences. In *Pinus sylvestris* pollen tubes, no acidic pectins were found [55]. Later, in some species, a certain amount of both esterified and acidic pectins was nevertheless found [58,59], but their pattern differs dramatically from the one typical for angiosperm pollen tubes: in Norway spruce (*Picea abies*), neutral pectins are present in the tube tip, and then disappear, sometimes remaining in trace amounts as stripes [45]; in Meyer spruce (*Picea meyeri*), on the contrary, acidic pectins were found along the entire length of the tube, including the tip, and esterified pectins were confined to the place of the tube emergence from the grain. Boron deficiency in the germination medium (B is involved in the formation of B-pectic gel) caused the accumulation of acidic pectins in the tip [59]. A critical actor in angiosperm pollen tube morphology is the wall-embedded enzyme pectin methylesterase (PME), which in type II PMEs is accompanied by a co-transcribed inhibitor, PMEI [60]. PMEs convert the esterified pectic primary wall to a stiffer state by pectin de-esterification. It was hypothesized that rapid and precise targeting of PME activity was gained with the origin of type II genes, which are derived and have only expanded since the origin of vascular plants [60]. Expression analysis with degenerate primers suggests type II PMEs are not expressed in the growing pollen tubes of *Pinus strobus* [61]. On the contrary, in basal flowering plant *Nymphaea odorata*, transcripts of four type II PME homologues and 16 type I PME homologues were found, which were more abundant in pollen grains and pollen tubes than in vegetative tissues [61]. It was assumed that improved control of PMEs, pollen-active enzymes that mediate de-esterification of pectins near the pollen tube tip, is a conservative feature of angiosperms that partly determines their ability to grow rapidly [61].

Cellulose is present at the pollen tube tip of *Pinus sylvestris* and *Picea abies*, and in the tube body becomes the main cell wall component [45,55]. Specific inhibition of cellulose deposition by isoxabene leads not only to growth arrest and tube swelling, but also to microtubule disorganization. [45]. The effect of isoxabene on spruce pollen tubes thus confirms the hypothesis of a mutual relationship between microtubules and cellulose in these cells [5].

## 5. Biochemical Regulation of Conifer Pollen Germination and Pollen Tube Growth

### 5.1. Cell Respiration

In gymnosperms, the activation process differs from the one in angiosperms, firstly, in speed, and secondly, in the presence of a carbohydrate reserve—starch, which can be accumulated and decomposed depending on the needs of the male gametophyte and the presence of sugars in the germination medium. Thus, respiration of pollen grains in mountain pine (*Pinus mugo*) is not divided into distinct phases, but increases linearly, starting from 2 h of incubation. Before this point, immediately after hydration, there is a short lag phase, during which oxygen uptake slowly increases [26]. Analysis of the ATP content showed that during this period there was a depletion of ATP reserves, apparently formed in the pollen grain before dehiscence. When oxygen uptake began, the ATP level was restored and then gradually increased until the pollen tube had appeared (about 16 h) [62]. It should be noted that pollen was incubated in a medium with a carbohydrate component, and thus it was able to synthesize starch. Later, data on the respiration of pine (*Pinus ponderosa)* pollen grain suspension were obtained by infrared spectroscopy: in the first 24 h of incubation, CO_2_ release was low, after which it increased rather sharply, the most active increase was observed after 48 h of incubation [63]. For blue spruce (*Picea pungens*), the pollen of which germinates faster than that of pine, oxygen consumption registered with the Clarke electrode increased sharply from 2 to 9 h of incubation and the maximum coincided with the pollen tube appearance; then, during early stages of pollen tube growth, the level of respiration decreased by about 30% [64]. In the same species, infrared spectroscopy showed that from 8 h of incubation, the release of carbon dioxide increased sharply and almost linearly [63]. These data are difficult to compare, since in the last study, starting at 8 h, respiration was measured once a day, and in the first one, the points of 9 and 14 h were compared. However, 8 and 9 h points, respectively, were marked as tipping points in the studies, which appeared to be associated with the moment of the tube emergence [64].

In some studies, two phases during conifer pollen germination were allocated for convenience, with phase I covering the first 12 h, the second—the next 12 h (time intervals for pine pollen) [65]. Conventionally, we can say that the first phase refers to non-germinated grains, and the second refers to growing tubes. Studying the effect of carbohydrates’ availability on respiration, the authors found that in the presence of optimal carbohydrates—sucrose or fructose—respiration in phase II is approximately two times higher than in phase I. In this case, not only ATP synthesis occurs, but also intense starch synthesis, which is a characteristic feature of conifer pollen grains [65].

Germination occurs slower in Douglas fir (*Pseudotsuga menziesii)* than in pine and spruce [28]. Respiration remained high and constant during the first 36 h, and then increased sharply (apparently due to the germination), while ATP content increased rapidly already during the first 8 h and remained high until the end of the observation period [28]. During the first 24 h in culture, ATP content increased by 86%, while the content of ADP and AMP decreased. It should be noted that, unlike pine, *Pseudotsuga* is able to support pollen tube growth without sugars in the medium [28]. Perhaps this is the reason for a relatively constant level of respiration. Other authors found that in this species there is a direct correlation between the level of pollen respiration and its fertility [66]. Thus, they proposed using respiration (and not germination, which takes a long time and requires sterile conditions) to quickly assess the quality of fir pollen in forest plantations [66]. The same correlation was found for *Pinus taeda* [67].

### 5.2. RNA and Protein Synthesis

One of the features of pollen germination in conifers is the active RNA synthesis. In many angiosperm species, pollen germination has been studied for the dependence on transcription and translation. It was found that mRNA required for germination and early tube elongation is already present in pollen grains at dehiscence [68]. The same applies to proteins required for germination. In most of the studied plants, cycloheximide (translation inhibitor) does not suppress pollen germination and pollen tube growth but prevents the formation of male gametes, apparently due to the lack of proteins required for division spindle formation [69,70,71]. Apparently, in most flowering plants the synthesis of new proteins is required only at the stage of pollen tube growth, but proteins for germination are being stored during gametophyte maturation. However, the situation in conifers is completely different: back in the 1960s, it was found that the nuclei of the generative cell and the tube cell in *Pinus ponderosa* pollen actively synthesize RNA at the early germination stages [72]. This conclusion was later confirmed in other pine species. For example, Fernando et al. analyzed RNA and protein synthesis in California mountain pine (*Pinus monticola*), as well as other conifers: four species of pine, two species of spruce, Himalayan cedar (*Cedrus deodara*), and Pacific silver fir (*Abies amabilis*) [73]. RNA and protein synthesis are important for pollen germination in all studied plants, as indicated by the effects of actinomycin D and cycloheximide, respectively. Pollen grains germinate in the presence of actinomycin D, but further tube elongation is suppressed. This suggests that the RNA required for germination is already available in mature pollen, but tube elongation depends on the synthesis of new RNA [73]. The latter is confirmed by experiments with inhibitors of histone deacetylase, which maintains individual chromatin regions in an inactive state [74]—changes in the enzyme activity caused numerous growth abnormalities, from a disturbed Ca^2+^ gradient to altered cell wall deposition [75]. These experiments demonstrated the active use of chromatin during pollen tube growth in Wilson spruce (*Picea wilsonii*) [75].

As for the translation, it was shown using cycloheximide for western white pine (*Pinus monticola*) that many proteins required for germination and tube elongation are not yet present in mature pollen. These proteins are apparently synthesized at the beginning of pollen germination and during the tube elongation. The effects of transcription and translation inhibition in eight other conifers were the same as in *P. monticola*, indicating a general trend [73].

Proteomic approach made it possible to detect and classify, albeit with some limitations, proteins synthesized in germinating pollen. Thus, a comparative analysis of the protein profiles in *Pinus strobus* pollen before and after germination revealed 57 differentially accumulated proteins (which was about 9% of all separated protein spots). They fall into the following functional groups: metabolism, stress/defense response, gene regulation, signal transduction, and cell wall formation [76]. We conclude that, despite the abundance of pollen proteomic studies, as well as a sufficient number of reviews [77,78,79,80], gymnosperms are still poorly studied in this respect.

### 5.3. Ion Homeostasis

It is generally accepted that the key role in pollen tube growth regulation in angiosperms is played by Ca^2+^, and it is clear that the role of this regulator in pollen germination in conifers is of great interest. Bunge pine (*Pinus bungeana*) [81] and different spruce species [44,53,82] were used as model objects in these studies. The distribution of Ca^2+^ in pollen tubes was studied with injected fluorescent dyes, which allowed for the revealing of apical calcium gradient, although not as steep as in angiosperms. In *Picea abies* pollen tubes, the cytosolic Ca^2+^ concentration decreased from 450 nM at the apex extremum to 225 nM at the base of the clear zone [53]. A similar pattern was observed in other species under control conditions [81,82].

Blocking plasmalemmal calcium channels, which naturally reduced calcium currents, caused gradient dissipation and decreased intracellular Ca^2+^ concentration in *Picea abies, P. wilsonii*, and *Pinus bungeana* pollen tubes [53,81,82]. In Bunge pine, inhibitor nifedipine caused a number of cytological effects and affected the set of expressed proteins [81]. An early response (within an hour after blocking calcium channels) included disturbances in organelles ultrastructure (mitochondria, Golgi apparatus, and ER) and changes in the composition of proteins involved in energy generation and signaling [81]. Following this (and obviously, as a result of this), actin depolymerization began, and an imbalance of endo- and exocytosis and disturbances of cell wall construction occurred. These results demonstrate the necessity of calcium inward fluxes for the comprehensive maintenance of active pollen tube growth. In Wilson spruce, the effects of calcium channel blocking on cell wall assembly were studied. As it was shown by two independent methods [82], in addition to the expected effects on growth, both inhibitors strongly affected the tube morphology, for example, causing callose accumulation in the tip region, as well as altering the deposition of other wall components, such as pectins and arabinogalactan proteins, which indicates a close relationship between calcium homeostasis and cell wall construction.

A similar approach was applied to reveal the role of Ca^2+^/calmodulin signaling in *Picea meyeri* pollen tube growth [44]. Using calmodulin antagonist trifluoperazine, about 90 proteins with calmodulin-controlled accumulation were identified. Inhibition of Ca^2+^/calmodulin signaling caused a number of alterations in the growing pollen tube, similar to those described above, including disturbances in organelles ultrastructure, actin depolymerization, disorganization of endocytosis and exocytosis, and modified cell wall composition. The dynamic study of protein profiles showed that the first to be affected are proteins associated with signaling, organelles functioning, and energy generation. Later changes affected cytoskeleton-related proteins, proteins of secretory pathways, and polysaccharide synthesis. Thus, it can be concluded that Ca^2+^/calmodulin signaling plays an important role in the regulation of conifer pollen tube growth. Later, one of the proteins involved in signal transduction from calcium to the cytoskeleton was described in Norway spruce [48]. A kinesin-like calmodulin-binding protein (KCBP), a member of the Kinesin-14 family, binds to calmodulin and regulates organelle motility in *Picea abies* tubes. PaKCBP is located in the growing tip where it colocalizes with microtubules. Inhibition of calmodulin by perfusion of its antagonist W-12 into the tubes reversibly slows down pollen tube growth, affects organelle motility, and promotes the formation of cytoskeleton bundles along with an increase in the PaKCBP level [48]. Thus, one of the regulatory chains leading from Ca^2+^ to the organelle motility in conifer pollen tubes is already clear in general (Figure 5).

In addition to calcium, protons and membrane potential are important for conifer pollen germination. The study of intracellular pH dynamics during pollen activation in blue spruce showed that both pH and membrane potential change after the first cytological signs of germination (namely, breaks in the exine) [64]: after 9 h of incubation, there is a significant cytoplasm alkalization and membrane hyperpolarization. Further, when the tube appears, the pH stabilizes, and membrane potential increases slightly (in modulus). The first change in ion homeostasis revealed to date is the release of anions from pollen grains; by the time exine ruptures appear, it has already been completed [64]. It is not yet clear how important these changes are, except for the activity of H^+^-ATPase: P-type ATPase inhibitor orthovanadate blocked pollen germination, indicating significance of the enzyme for germination.

In *Picea pungens* pollen tubes, gradients of pH and membrane potential were found, which differed in shape from the corresponding gradients previously reported for flowering plants. Plasmalemma in the apical part, similarly, is depolarized [83], but in lily (*Lilium longiflorum)* and tobacco (*Nicotiana tabacum*), the difference in membrane potential values, revealed by the same method, was found to be much larger [84,85]. A variety of ion transport systems, such as Ca^2+^- and K^+^-conducting channels, anion channels, and H^+^-ATPase, are involved in maintaining the gradient, according to inhibitory analysis [83]. Proton pump is also responsible for pH gradient in pollen tube cytoplasm [64]. The highest pH values were observed in the tube apex (Figure 5), which corresponds to the proton pump location on the plasmalemma and distinguishes the spruce from angiosperm pollen tubes, in which the alkaline zone was located in the subapex, and H^+^-ATPase was predominantly localized there [86,87,88]. Hypothetically, the discovered feature can be associated with the typical distribution of organelles in the cytoplasm, cytoskeleton structure, and the steepness of Ca^2+^ gradient, which also distinguish conifer pollen tubes from the tubes of flowering plants (Figure 5).

### 5.4. Reactive Oxygen Species (ROS) and NO

Redox homeostasis is one of the main regulatory systems during pollen germination and tube growth in flowering plants, and it has been actively studied in recent years, especially regarding pollination and pollen behavior in vivo [89,90,91,92]. However, much less is known about the role of this system in gymnosperms [89]. For example, ROS can affect the germination pattern in *Picea pungens*, altering rigidity of the cell wall and its resistance to rupture [31]. Similar data were obtained for tobacco, but in that case, germination efficiency and cell wall resistance to osmotic shock were altered, while in spruce, under optimal conditions, a significant part of the grains produces two tubes instead of one, and the effect of ROS on the wall was found to block this process [31]. On the other hand, exposure to hydrogen peroxide did not affect germination efficiency in spruce, although tube growth was accelerated in the presence of moderate and even rather high peroxide concentrations [83]. Compared to angiosperms, the picture was fundamentally different: there, low peroxide concentrations stimulated germination, while high concentrations inhibited it [93]. On the other hand, quenching endogenous ROS or blocking NADPH oxidase activity by DPI (diphenyleneiodonium chloride) in spruce and cypress led to a sharp decline in the germination efficiency and growth speed [83,94]. In addition to germination and growth inhibition under the influence of DPI, the Arizona cypress (*Cupressus arizonica*) also showed severe violations of the tube morphology and cytoskeleton structure [94]. Thus, we can conclude that, at the germination stage, pollen of coniferous plants does not expect an exogenous influence by ROS, in contrast to flowering plants, in which ROS are used for communication between the pistil and the male gametophyte. At the same time, endogenous ROS are generated in spruce pollen, which was recorded very early during the activation [83] and ROS, apparently, are necessary for growth processes.

The polar distribution of ROS in gymnosperm pollen tube has been described in several species, including Meyer spruce (*Picea meyeri)* [95] and cypress [94] (Figure 5). In *P. meyeri*, local ROS production at the tube tip was mediated by NADPH oxidase and was associated with lipid microdomains, which was shown by specific staining and immunolocalization [95]: isolating sterols using filipin, the authors recorded smoothing of both apical ROS and Ca^2+^ gradient, which, among other findings, indicates their relationship. In vitro enzyme activity was also sterol-dependent [95]. Later, using specific dyes, mapping of hydrogen peroxide and mitochondrial superoxide radical was carried out in short pollen tube initials and tubes. A pronounced accumulation of H_2_O_2_ was found in the apical cytoplasm of *Picea pungens* pollen tubes, superoxide radical is localized in mitochondria, and peroxide is also concentrated in amyloplasts [83]. In general, similar distribution is typical for flowering plants, for example, for lily [96]. 

Are there systems in conifer pollen that not only produce ROS but also control their concentration and ratio? Thus far, this issue has not been studied enough. To date, it has been reported that ascorbate peroxidase—an enzyme capable of inactivating hydrogen peroxide and hydroxyl radical—is present in *Picea meyeri* and *Pinus strobus* pollen proteome [44,54,76] during germination; moreover, in pine, this protein appears only after germination and is absent in mature pollen [76].

The first studies of NO in gymnosperm pollen germination gave an unexpected result. If angiosperm pollen tubes respond to NO by slowing down and changing growth direction, demonstrating an avoidance response [97,98], the Bunge pine tubes, on the contrary, accelerate their growth [99]. NO donor stimulated Ca^2+^ entry into the pollen tube and increased the steepness of apical calcium gradient; it also influenced actin organization and vesicular transport. Inhibition of NO synthase or inactivation of NO had the opposite effect: a decrease in intracellular NO level affected cell wall structure and composition, while pollen tube growth stopped. Thus, in pine, NO controls the configuration and distribution of cell wall components and also affects transmembrane calcium fluxes and F-actin organization [99]. Similar data were obtained for cypress: an inhibitor of NO synthesis and its quencher also suppressed the tube growth, causing morphological abnormalities and destroying the cytoskeleton [94].

As for NO distribution, it also differs in pollen tubes of conifers and flowering plants. Thus, in cypress, NO accumulation was observed in the nucleus and in the apical part, similar to ROS [94], while in *Lilium longiflorum*, such an accumulation was absent, and all NO was localized in peroxisomes, which are located quite far from the apex, and was not detected in the cytoplasm [97]. The role of NO in the regulation of pollen tube growth in conifers remains to be studied.

Another molecule with a potential regulatory significance is ATP (apoplastic forms are referred to as aATP). Using bioluminescence analysis, Zhou et al. [100] found that *Picea meyeri* pollen grains release ATP into the extracellular matrix prior to germination and during the tube elongation. The addition of exogenous ATP or an inhibitor of apyrase (an enzyme that hydrolyzes aATP) to the pollen suspension inhibited germination and elongation of pollen tubes, disorganizing the microfilaments structure [100]. On the other hand, exogenous apyrase also had an inhibitory effect, as did purinoceptor inhibitors. ATP increased Ca^2+^ influx during germination, which was inhibited by purinoceptor inhibitors. Thus, the authors put forward the concept of the optimal concentration of aATP, which is of key importance for the initiation of spruce pollen germination, as well as for Ca^2+^ homeostasis in growing pollen tubes [100]. This demonstrates a similarity with angiosperms: exogenous ATP, but not ADP, suppressed both pollen germination and tube growth in *Arabidopsis*, and the same effect was exerted by the suppression of the activity of apyrases, APY1 and APY2 [101].

## 6. Future Perspectives

One of the promising directions for this topic is the deepening understanding of the regulatory mechanisms that differ in gymnosperms and angiosperms. In this sense, pine or spruce pollen suspension is a very convenient model system, as it is rather well studied in comparison to the representatives of other taxa, and, accordingly, the integration of new additional knowledge will help build an adequate model of the origin, development, and transformation of regulatory systems on the way from conifers to flowering plants.

Another interesting direction is specialization, that is, clarification of the features of pollen physiology in special groups of gymnosperms, and in this sense, cycads, Gnetales, and ginkgo are of great interest, since the pollination process in these plants has been studied from the point of view of morphology and anatomy, but the data on pollen physiology are missing, and very little is known about pollen tube growth regulation and the peculiarities of its functioning as a haploid organism.

## Figures and Tables

**Figure 1 plants-10-01301-f001:**
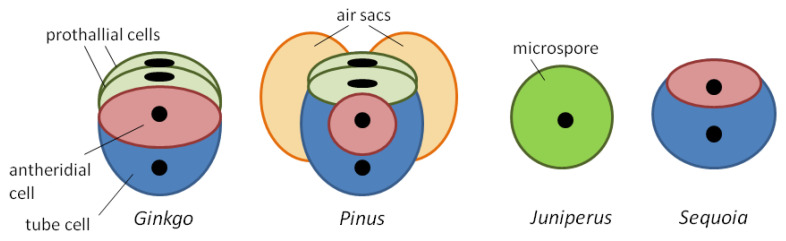
Cellular composition of some gymnosperm pollen grains at dehiscence.

**Figure 2 plants-10-01301-f002:**
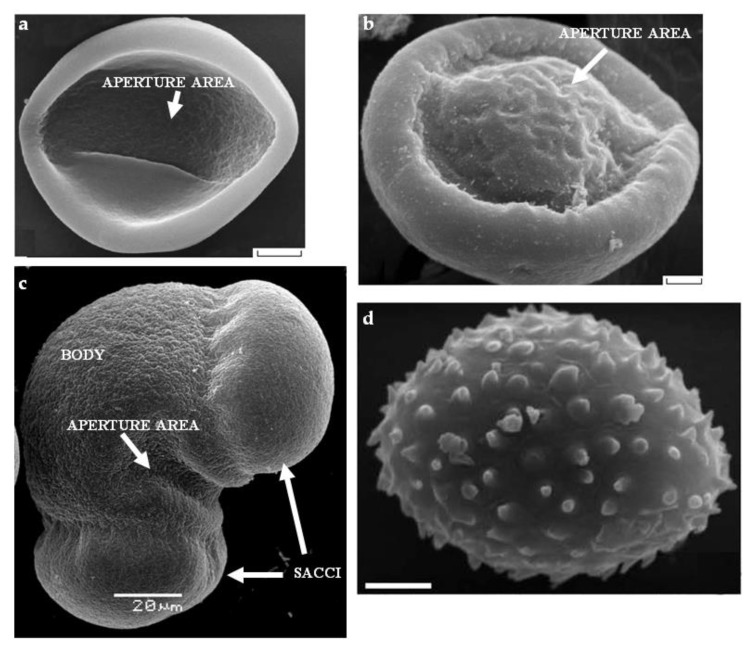
SEM of gymnosperm pollen: (**a**) *Cycas micholitzii* Dyer (Cycadaceae); (**b**) *Ginkgo biloba* L. with a bulge in the aperture area; (**c**) *Picea pungens* Engelm. with typical saccate morphology; (**d**) *Gnetum macrostachyum* Hook with microechinate sculpture. Scale bar: (**a**,**b**,**d**)—3 µm. Pictures are from the following articles: (**a**,**b**) [9], (**d**) [10].

**Figure 3 plants-10-01301-f003:**
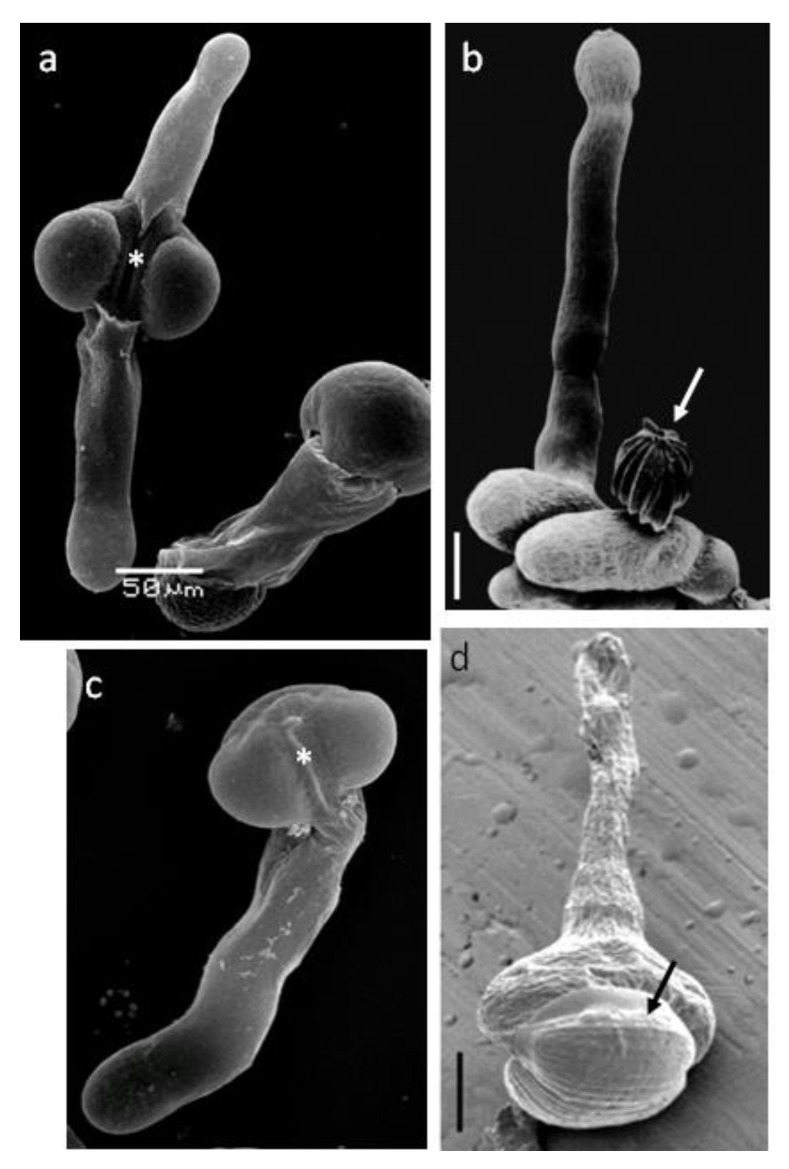
SEM of in vitro pollen germination in three gymnosperm species: *Picea pungens* Engelm. (**a**,**c**), *Ephedra americana* Humb. & Bonpl. (**b**), and *Welwitschia mirabilis* Hook. f. (**d**): unipolar (**c**) and bipolar pollen germination in *Picea* (**a**) with exine covering the pollen grain, aperture furrow marked by asterics; pollen tube in *Ephedra* with the exine shed (marked by an arrow) (**b**); growing pollen tube with the grain partly covered by the exine (marked by an arrow) (**d**). Scale bar: (**a**)—50 µm (the same magnification in (**c**)), (**b**)—20 µm, (**d**)—15 µm. Pictures are from the following articles: (**a**,**c**) [31], (**b**) [32], (**d**) [35].

**Figure 4 plants-10-01301-f004:**
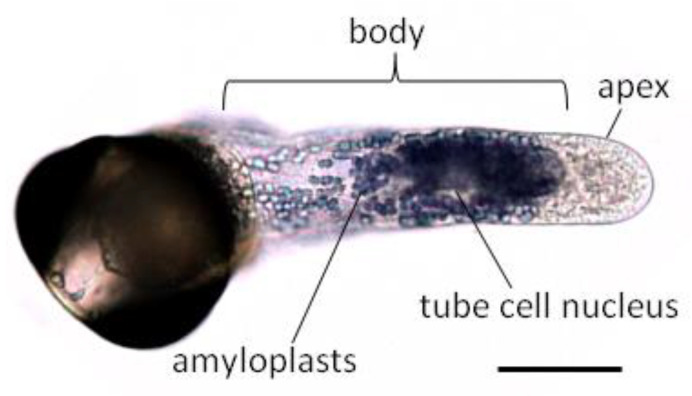
*Picea* pollen tube after 20 h of incubation in vitro. J_2_ + KJ starch-specific staining. Bar—50 µm.

**Figure 5 plants-10-01301-f005:**
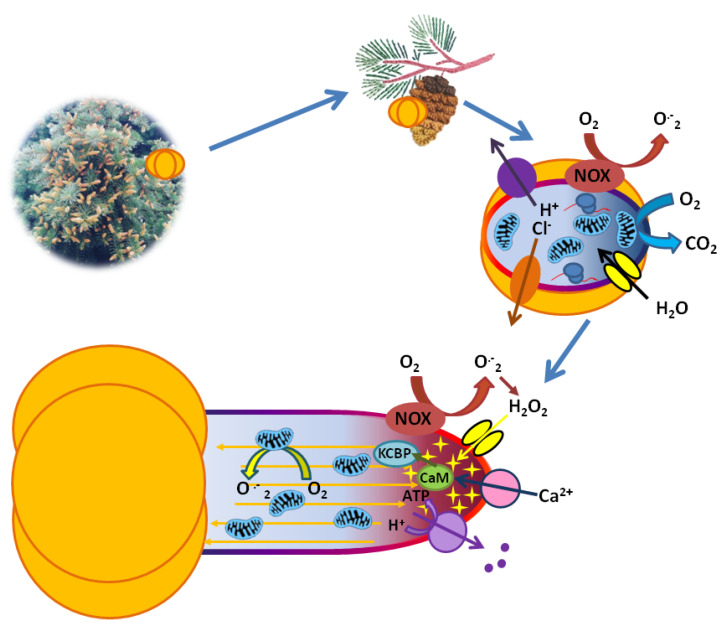
Structural and regulatory network in germinating conifer pollen and growing pollen tube. Mature pollen travels from male cones to female cones by wind, highly dehydrated and dormant. After its landing on a pollination drop, rehydration and then physiological activation occur. Anions are released from the grain (orange—putative anion channels involved), ROS are produced by NADPH-oxidase (NOX, brown), respiration is activated (blue), protein synthesis occurs in ribosomes, pH shifts from acidic to slightly alkaline values (cytoplasm gradient staining from left to right = from early to later activation stage) due to H^+^-ATPase activity (purple), and membrane hyperpolarization occurs (membrane gradient staining, from left to right).

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
