# Peer review of "Pollen Germination and Pollen Tube Growth in Gymnosperms"

_plants, 2021, doi:10.3390/plants10071301_

Round 1
Reviewer 1 Report
This manuscript is dedicated to a relatively weakly characterized topic and hence deserves publication. There are only minor (although abundant) suggestions and corrections concerning style and language. For example, authors are strongly recommended to unify a mode to call plant species, i.e. if Latin or English names are presented. All my suggestions can be found in a manuscript file (see attached).

Author Response
Many thanks for the thorough analysis of our manuscript, including both semantic advice and stylistic editing. All your comments have been taken into account in the revised version of the text. Unfortunately, the number of revisions after three reviewers was so great, including complete changes to some sections and the addition of new figures, that it is not possible to highlight a particular revision in color. But we believe that thanks to your comments, the text has improved significantly.
Reviewer 2 Report
Currently, majority of pollen research is focus on angiosperms. However, research on pollen from gymnosperms should not be ignored, and it’s important to thoroughly review what have been found to date. The manuscript entitled “Pollen germination and pollen tube growth in gymnosperms” mainly reviewed physiological studies on conifer pollen, but also briefly described features of gymnosperm pollen grains, and pollen germination in gymnosperms. The review after a careful revision would be a valuable contribution to the field.
Major comments:
- Some conclusion or statements are misstated, overstated or without supporting evidence. See below for some of them.
- Some references are missing to support the statements. See below.
- For Figure 2, since this is a review about pollen germination and tube growth in gymnosperms, the authors should gather representative germination images from different (representative) gymnosperms, rather than just from 1 species (Picea pungens). The corresponding context also needs to be better expanded.
Minor comments:
Fig 1, use arrows or so to point out aperture, exine and intine etc in all figures.
Line 63, what do you mean by “the relief of the exine is not expressed” and “exine is relief” in line 65?
Line 64-65, “In Gnetales pollen can be apertureless with internal polarity or monosulcate … [10,15,16].” The authors need to be carefully describing the aperture number of Gnetopsida. Ref 15 described pollen from Ephedra and Welwitschia, both of which belong to Gnetopsida. For Welwitschia, “Pollen grains of Welwitschia mirabilis… are monosulcate” (page 8 in ref 15). However, for Ephedra, they claimed that the aperture condition was still not clear although they described pseudosulcus, see “The aperture condition in Ephedra is unclear and has been variably described as polyaperturate … and inaperturate ... Between each ridge a furrow is located ... This furrow, which can be either branched or unbranched, will here be called the pseudosulcus, as suggested by Huynh (1975) ” in page 2 of ref 15.
Line 69-71, “The sacs are located at the distal end of the pollen grain … the floating of pollen grains (because of the sacs?) in pollinating droplets … is an effective mechanism for their retraction into micropyle; therefore, air sacs serve not only for pollen movement by the wind [17].” The logic is not clear here. Air sacs are not introduced ahead. Also, need to briefly introduce what is pollinating droplets (eg, ovular secret exudation) for broader audience (move the introduction info from line 106 here).
Line 91-92,” The latter (flowering plants) in most cases have an aperture - area not covered with exine, intended for the fast tube outlet [23].” 1) the definition of aperture needs to be introduced earlier. 2) The predominance aperture number is not correctly described for flowering plants; see “Furness and Rudall. 2004. Pollen aperture evolution—A crucial factor for eudicot success? Trends in Plant Science 9: 154–158. doi:10.1016/j.tplants.2004.01.001”, “Aperture number influences pollen survival in Arabidopsis mutants (2016), doi:10.3732/ajb.1500301” or “Effect of aperture number on pollen germination, survival and reproductive success in Arabidopsis thaliana (2018), doi: 10.1093/aob/mcx206”. 3) ref 23 is not appropriate for this part as ref 23 is mainly focus on gymnosperms pollen.
Line 111-112, the work from ref25 or others analyzing chemical compound of pollination droplet should also be cited.
Line 126-127, “In some Tsuga species and all Araucariaceae species” do they produce pollination droplet?
Section 3 “Pollination and pollen germination in gymnosperm species”, better to reorganize the content according to the findings from in vitro germination and in vivo germination (here it read like: in vitro germination, in vivo germination, and then back to in vitro germination again).
Line 150, “Protocols have been developed for many species”, list the corresponding references.
Section 4.1, references supporting several statements are missing. The authors need to show the references for any information that is cited or concluded from other work. Same comments for the following sections.
Line 177, “microtubules are not a necessary participant in polar growth …” the authors need to be carefully making this conclusion as there are data suggesting a role of MTs in PT tip growth.
Line 180-183, “a similar effect was observed in experiments with modulation of intracellular [Ca2+] in pollen tubes [45]. From these data, it can be concluded that … which, in turn, are controlled by Ca2+ (Figure 3).” The authors cannot get the latter conclusion based on the former information. The authors need to collect more evidence or to be careful in statements.
Line 210-211:” angiosperm pollen … consist of two layers: the pectin outer layer and the callose inner layer” It’s not accurate in describing PT wall from angiosperms. There is a singly wall layer in the pollen tube tip and two layers in the shank. And, although simple names were given to the two layers in the shank, other cell wall components are also present and tightly crosslinked with pectin or callose. Additionally, the authors need to be careful to claim that “none” of the studied gymnosperm pollen tubes had wall similarity to that of angiosperm pollen as if you look at things from a different angle, there are similarities.
Section 4.2, need a better transition in the second paragraph (line 215).
Line 219-223, “pollen-specific pectin methylesterases were not found in basal angiosperms or gymnosperms [50]. When studying the expression of PME genes in a basal flowering plant Nymphaea odorata, transcripts of four type II PME homologues and 16 type I PME homologues were found …”. Need to explain the difference between pollen specific PEM and type I/ II homologues.
Line 236, need to explain what “two-way relationship” is.
Section 5.1 and 5.2: The authors need to point out that this part of review is mainly based on findings/data from very old publications; updates with modern technology are required/encouraged for following up etc.
Line 356, “regulates intracellular motility”, motility of what? Line 357-358, “slows down growth”, growth of what? Please make the meaning of the sentences clear.
Figure 3, 1) the colors (orange and purple/blue) reflecting the acidic/alkaline status in the pollen grain and the growing pollen tube seems to be opposite. 2) Any updates on calcium homeostasis in pollen grains before germination?
Author Response
Many thanks for the thorough analysis of our manuscript, including both semantic advice and stylistic editing. Unfortunately, the number of revisions after three reviewers was so great, including completely changing some sections of the text and adding new figures, that it is not possible to highlight a specific revision in color. But we believe that thanks to your comments, the text has improved significantly.
I will answer those comments that I could not use to the fullest.
Section 5.1 and 5.2: The authors need to point out that this part of review is mainly based on findings/data from very old publications; updates with modern technology are required/encouraged for following up etc.
Indeed, here I am referring to old data, but this is due to the fact that the study of this topic was suspended. Since the 90s, respiration of pollen grains or the total synthesis of RNA and protein has been studied mainly in connection with practical issues (for example, the assessment of the quality of pollen for agriculture by the level of respiration). I included a couple of such works on gymnosperms in this review, but, in general, there are almost no new articles on these topics. Maybe I'm wrong, but after receiving the review, I looked for them again. I added an interesting piece of work from the 90s, which I did not find the first time, but it can hardly be called "modern data". However, I believe that the lack of new data is not a reason to discard these sections from the review. On the contrary, I wanted to emphasize that little is known about these fundamental processes. Maybe this will push someone to conduct such studies on rare species or plants from different taxa.
Figure 3, 1) the colors (orange and purple/blue) reflecting the acidic/alkaline status in the pollen grain and the growing pollen tube seems to be opposite. 2) Any updates on calcium homeostasis in pollen grains before germination?
- In the pollen grain, the blue gradient reflects dynamic pH changes (alkalinization), and in the tube it shows spatial heterogeneity (alkaline tip which was reported for spruce). I tried to reflect the difference in the caption.
- I found studies on calcium in the developing male gametophyte, but we decided to leave the development of microspores and pollen outside the scope of this review. And in a mature, non-germinated pollen grain I did not find anything new; there are very few works on gymnosperms, and on flowering ones - almost everything is on tubes, except for the fact that calcium in the medium is needed for germination.
All your comments have been taken into account in the revised version of the text. Thanks again.
Reviewer 3 Report
Very interesting review article. Congrats to authors.
Just some minor corrections are suggested. All are listed below with an appropriate Line number(s) from text in order to facilitate tracking:
1. Maybe in Introduction section it should be defined, in more precise way, differences between terms gymnosperms and conifers since you intend to talk about the first one and than in Line 31 you are start to talk just about conifers pollen.
2. One general question for Introduction section: do pollen grains of gymnosperms have intine membranr? Because it does not than it is also immportant difference from angyosperms and it should be mentioned in Introduction section.
Line 79: Should be "Gnetum" written in Italic since it is plant genus?
Line 81: Suggest to reorder as follow: "Cycadales and Gnatales species"
Line 105: Find some synonim for pollination in order to avoid repeatining of the same term.
Line 110: Where it is studied, give some reference here?
Lines 119 and 120: Pinus and Picea in Italic maybe?
Line 147: The same as previous.
Line 183: Add "ions" after "Ca2+".
Line 185: Put Latin plant name in Italic here.
Line 200: To my best knowledge callose also disapear in pollen of angyosperms with maturation. So, differences than are related more to localisation of calose not to maturation. Please check/clarify.
Line 318: Add "ions" after "Ca2+". Aply this through a whole subsection 5.3.
Line 419: Define DPI abbreviation.
Author Response
Thank you very much for your thorough analysis of our manuscript and your positive feedback. Unfortunately, the number of revisions after three reviewers was so great, including completely changing some sections of the text and adding new figures, that it is not possible to highlight a specific revision in color. I will answer some of the comments, the rest we just took into account when revising the text of the review.
- One general question for Introduction section: do pollen grains of gymnosperms have intine membrane? Because it does not than it is also important difference from angiosperms and it should be mentioned in Introduction section.
As far as I understood, we are talking about the inner layer of the pollen wall - intine. Gymnosperms also have it, and there are no fundamental structural differences between gymnosperms and angiosperms. There are some subtle differences, but to explain them, it is necessary to introduce additional technical terminology, which seemed to me excessive for this review, designed for a wide range of readers.
Line 318: Add "ions" after "Ca2+". Aply this through a whole subsection 5.3.
I looked through the articles of different authors and found that usually Ca2 + already refers to ions, since the charge is indicated. If written in words, then the word "ions" is added, for example, "calcium ions".
Line 200: To my best knowledge callose also disapear in pollen of angyosperms with maturation. So, differences than are related more to localisation of calose not to maturation. Please check/clarify.
One important function of callose, indeed, concerns the development of microspores, and this callose disappears during their maturation. Callose performs this function in both flowering and conifers, as far as I know. But we did not initially plan to consider the development of the male gametophyte in this review. Therefore, we do not discuss "that" callose. As for the callose, which appears later, in the pollen tube, here gymnosperms and flowering plants have differences, which I described here as best I could.